# Cell-free DNA as a potential biomarker of differentiation and toxicity in cardiac organoids

Brian Silver[1,2], Kevin Gerrish[2], Erik Tokar[1]*

[1]Mechanistic Toxicology Branch, DNTP, Research Triangle Park, United States; [2]Molecular Genomics Core, DIR NIEHS, Research Triangle Park, United States

**Abstract** Cell-free DNA (cfDNA) present in the bloodstream or other bodily fluids holds potential as a noninvasive diagnostic for early disease detection. However, it remains unclear what cfDNA markers might be produced in response to specific tissue-level events. Organoid systems present a tractable and efficient method for screening cfDNA markers. However, research investigating the release of cfDNA from organoids is limited. Here, we present a scalable method for high-throughput screening of cfDNA from cardiac organoids. We demonstrate that cfDNA is recoverable from cardiac organoids, and that cfDNA release is the highest early in differentiation. Intriguingly, we observed that the fraction of cell-free mitochondrial DNA appeared to decrease as the organoids developed, suggesting a possible signature of cardiac organoid maturation, or other cardiac growth-related tissue-level events. We also observe alterations in the prevalence of specific genomic regions in cardiac organoid-derived cfDNA at different timepoints during growth. In addition, we identify cfDNA markers that were increased upon addition of cardiotoxic drugs, prior to the onset of tissue demise. Together, these results indicate that cardiac organoids may be a useful system towards the identification of candidate predictive cfDNA markers of cardiac tissue development and demise.

*For correspondence:
erik.tokar@nih.gov

Competing interest: The authors declare that no competing interests exist.

## Editor's evaluation

This important study presents a comprehensive investigation into cell-free DNA (cfDNA) within the context of 3D cell cultures, offering valuable insights into this emerging research area. Despite being an exploratory study, the findings provide compelling evidence that continued progress holds the potential for the establishment of versatile 3D cell culture cfDNA assays. Such assays could serve as invaluable research and clinical tools, enabling monitoring of organoid growth and development, enhanced characterization of tissue dynamics, and ultimately facilitating the identification of novel biomarkers.

## Introduction

Biomarkers in blood or bodily fluids are valued for their potential as noninvasive diagnostics for early disease detection (*Zukowski et al., 2020*; *Leal et al., 2020*). Biological fluids contain many forms of information. One such parameter is cell-free DNA (cfDNA), which refers to free-floating DNA in the bloodstream or extracellular media (*Grabuschnig et al., 2020*). Although cfDNA is believed to be generated largely by apoptotic cells (*Grabuschnig et al., 2020*), this phenomenon is not limited to cells undergoing a death pathway. Active release of nucleic acids encapsulated in extracellular vesicles (*Kustanovich et al., 2019*) or through mitochondrial channels (*Xian et al., 2022*) are additional potential routes that may permit the release of cfDNA. Several measures, including concentration, fragment size, sequence, and epigenetic modifications such as differential

methylation, histone modification, and nucleosome spacing, can be used to characterize cfDNA and imply information about the cells from which it originated (*Zukowski et al., 2020*; *Lehmann-Werman et al., 2016*; *Moss et al., 2018*). Currently, cfDNA already has some diagnostic uses, including noninvasive prenatal testing for chromosome abnormalities (*Willems et al., 2014*), fetal gender determination (*Jacobsen et al., 2018*), and cfDNA-based tests for the detection of mutations and cancer screening (*Cisneros-Villanueva et al., 2022*; *Bronkhorst et al., 2019*). Although the clinical utility and accuracy of these tests in the prediction of cancer is uncertain in some cases (*Hackshaw et al., 2022*), clinical trials are promising and some FDA-approved tests are already available (*Cisneros-Villanueva et al., 2022*). Still, we have likely yet to unlock the full spectrum of clinical usage for cfDNA in noninvasive diagnostics. It remains unclear which specific events at the cellular level can be detected in cfDNA and what cfDNA profiles signify tissue-level changes during development and differentiation of specific tissues. Broadening our understanding of how cfDNA changes in response to normal tissue development and maintenance is necessary to strengthen recognition of aberrant cfDNA signatures that could reflect deleterious phenotypes such as teratogenesis or toxicity.

Human organoids have many advantages as models of both development and disease, including similarity to human tissues, applications in precision medicine, and high-throughput capabilities (*Kim et al., 2020*). Organoids are more complex than 2D cell cultures or spheroids, yet permit a more isolated view of specific tissues as compared to in vivo settings or human blood samples, which contain cfDNA from numerous tissue sources (*Moss et al., 2018*). These properties make human organoids an appealing system to study cfDNA in response to cell differentiation or chemical exposure. Although cfDNA has been successfully isolated from pancreatic organoids (*Dantes et al., 2020*), whether cfDNA can be recovered from additional organoid types is unclear.

In this study, we used cardiac organoid models to identify properties of cfDNA that were reflective of tissue differentiation and toxicity. Our analyses demonstrate the tractability of cardiac organoid systems to rapidly identify cfDNA markers associated with tissue-level events in a human-relevant setting. Further, we identified specific cfDNA sequences that were elevated prior to major tissue defects initiated by toxicant treatment, suggesting that cardiac organoids have potential value as a tool to explore cfDNA biomarkers of cardiotoxic events.

## Results

### Cardiac organoids derived from H9 embryonic stem cells exhibit characteristic morphological changes and express markers of differentiation

H9 embryonic stem cells were seeded, coalesced into spheroids, and induced to differentiate towards cardiac lineage (*Figure 1A*). The organoids began contracting rhythmically at approximately 20–30 beats per minute (bpm) on day 6 and maintained this beating throughout maturation (*Figure 1B*). We observed that the organoids expressed increased levels of several markers of cardiac differentiation at both the transcript (Troponin T, Nkx2.5) and protein (MEF2C, GATA-6, α-actinin, and Nkx2.5) levels, and decreased transcription of the pluripotency marker Oct 3/4 (*Figure 1C–F*).

### Cardiac organoids release cfDNA during their development

Conditioned media was collected from cardiac organoids at several time points during their maturation (*Figure 2A*). We observed that cfDNA could be extracted consistently during development of both organoid models, and total cfDNA concentration increased as the organoids matured (*Figure 2B*). To account for changes in cell number during organoid growth, we normalized the concentrations of cfDNA to levels of genomic DNA (gDNA) collected on the same growth days. Normalized cfDNA was the highest early in development (*Figure 2C*), indicating that cfDNA output may be higher in tissues on growth day 1, which are comprised of less differentiated cells. The recovered cfDNA from mature cardiac organoids on day 9 consisted of a broad distribution of fragments ranging from approximately 200–6000 base pairs (*Figure 2D*). Fragment distributions were more difficult to resolve at timepoints prior to day 6, likely due to low DNA concentrations and sensitivity limitations (*Figure 2—figure supplement 1*).

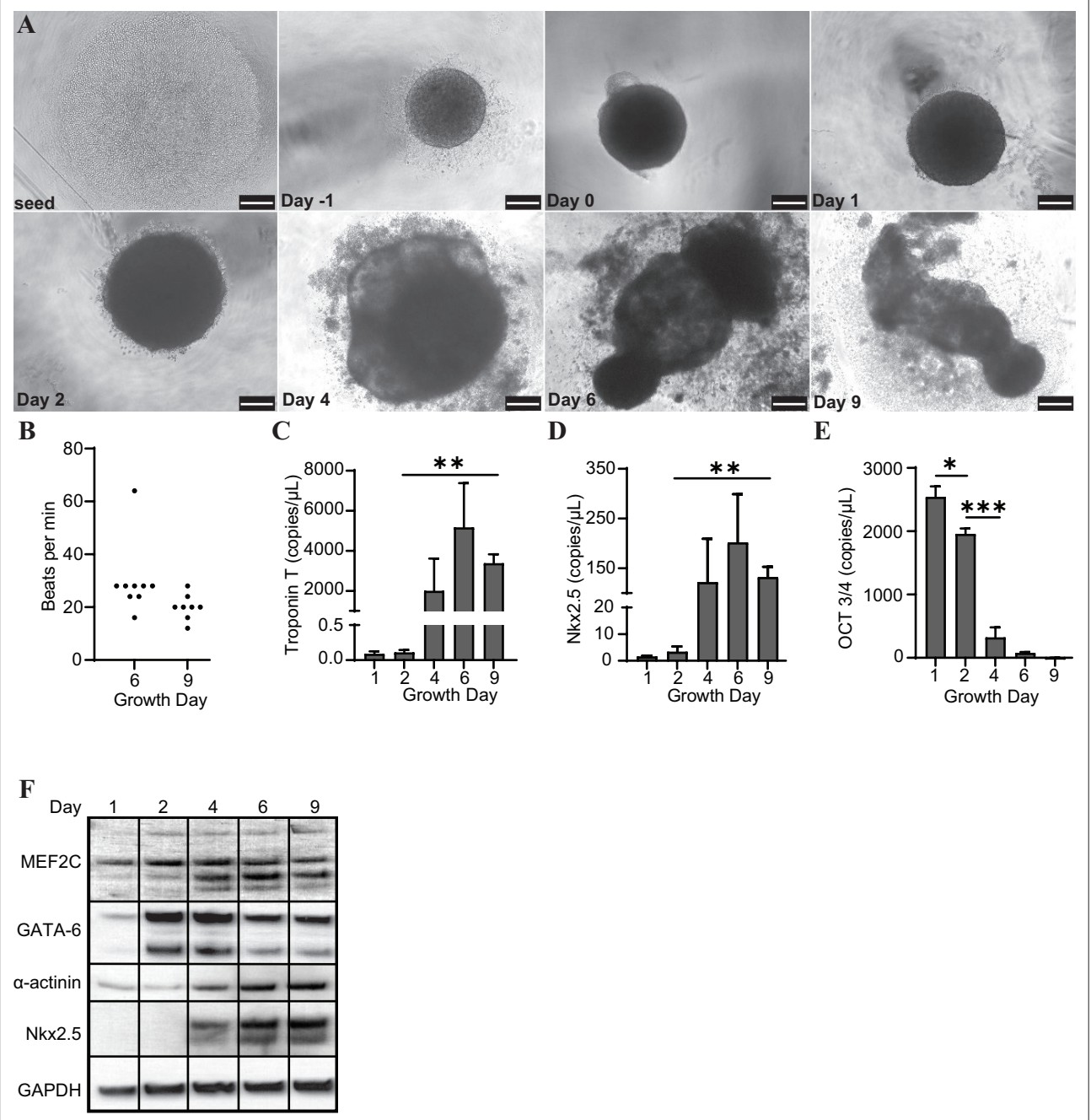

**Figure 1.** Cardiac organoids derived from H9 embryonic stem cells exhibit characteristic morphological changes and express markers of differentiation. (**A**) Brightfield images of cardiac organoids during growth and differentiation. Scale bars represent 100 μm. (**B**) Quantification of beats per minute across three separate biological replicates for growth day 6 (n = 9 organoids total) and day 9 (n = 7 organoids total). ddPCR of Troponin T (**C**), Nkx2.5 (**D**), and Oct3/4 (**E**) in cardiac organoids at different timepoints during differentiation, normalized to GAPDH. (**F**) Western blot showing protein expression of MEF2C, GATA-6, α-actinin, and Nkx2.5 relative to GAPDH in cardiac organoids at different timepoints. Graphs show average concentrations + SD. Samples were compared using an unpaired *t*-test with Welch's correction. *p<0.05; **p<0.01; ***p<0.001; ns, nonsignificant.

The online version of this article includes the following source data for figure 1:

**Source data 1.** Cardiac organoids derived from H9 embryonic stem cells exhibit characteristic morphological changes and express markers of differentiation.

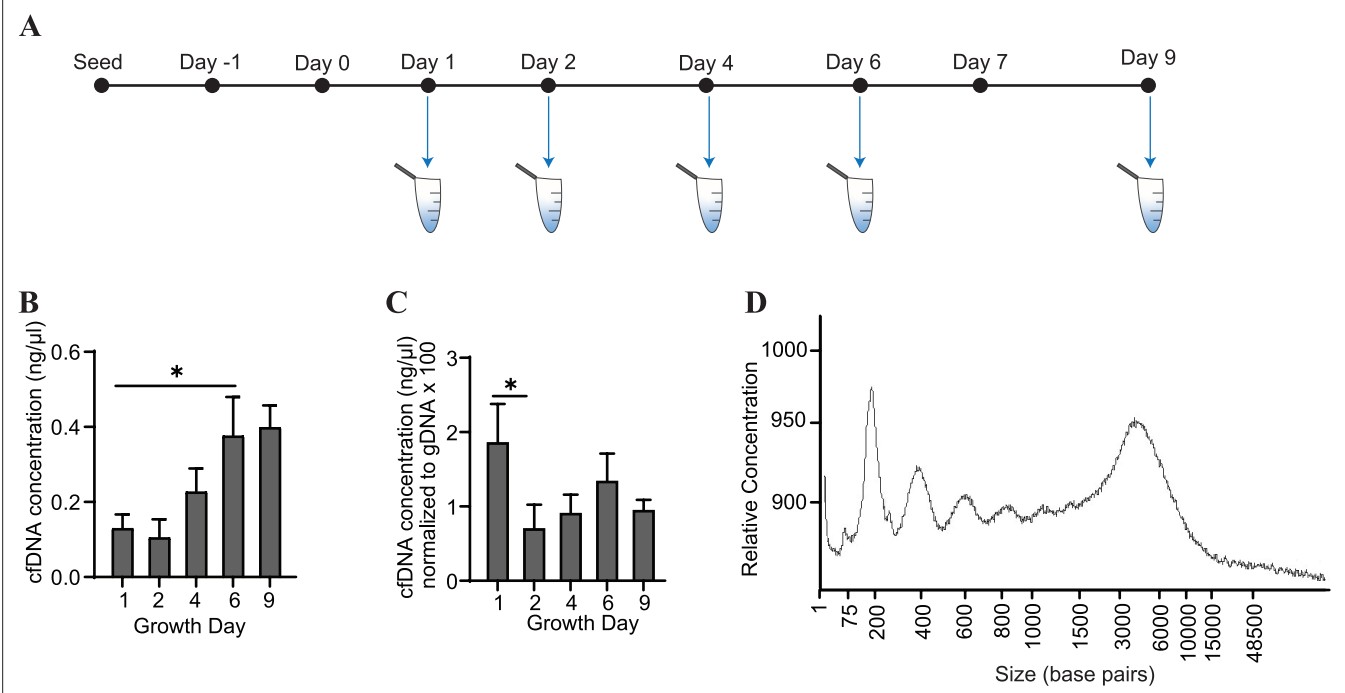

**Figure 2.** Cardiac organoids release cfDNA during their development. (**A**) Schematic illustrating timepoints of cfDNA collection in cardiac organoids. (**B**) Shown are cfDNA concentrations taken from n = 3 biological replicates during cardiac organoid differentiation. Data points represent the average of 2–3 technical replicates per biological replicate. (**C**) Cardiac organoid-derived cfDNA concentrations from panel (**B**), normalized to gDNA concentrations (multiplied by 100 to facilitate axis readability). (**D**) Electropherogram showing fragment sizes of cfDNA derived from mature cardiac organoids on day 9. Graphs show average + SD. Samples were compared using an unpaired *t*-test with Welch's correction. *p<0.05; **p<0.01; ***p<0.001; ns, nonsignificant.

The online version of this article includes the following source data and figure supplement(s) for figure 2:

**Source data 1.** Electropherogram showing fragment lengths of cfDNA derived from cardiac organoids at different time points during development.

**Figure supplement 1.** Overlay of representative electropherograms showing fragment sizes of cfDNA derived from cardiac organoids on different growth days.

**Figure supplement 1—source data 1.** Electropherogram showing fragment lengths of cfDNA derived from cardiac organoids at different time points during development.

## Cardiac organoids exhibit a time-dependent decrease in cell-free mitochondrial DNA abundance during growth

To further characterize the cfDNA recovered from cardiac organoids, we examined abundance of cell-free mitochondrial DNAcell-free mitochondrial DNA by quantifying the levels of markers located at several regions within the mitochondrial genome (*Figure 3A*). We observed that cardiac cfDNA levels of the mitochondrial genes ND1, mtCOX2, ND6, and ND4 dropped substantially in a time-dependent manner as the organoids differentiated (*Figure 3B*), and the markers were approximately equally represented in the cfDNA at most time points (*Figure 3C*). The tissue-level expression of mitochondrial genes slightly decreased midway through differentiation (*Figure 3D*), but did not fully account for the time-dependent drop in cell-free mitochondrial DNA. As anticipated, an increase in the mitochondrial protein VDAC/Porin was observed in mature cardiac organoids (*Figure 3E and F*), which is representative of increased mitochondrial size expected in cardiac tissues (*Guo and Pu, 2020*).

We hypothesized that mitochondrial fragments within cell debris might be expelled from the organoids as they differentiate, leading to higher cell-free mitochondrial DNA early in the growth process. To investigate this possibility, we examined cell-free mitochondrial DNA abundance in media conditioned by cardiac organoids that had been ultracentrifuged (100,000 × *g*) for 1 hr, to eliminate small fragments of cellular debris and extracellular vesicles. We observed similar cfDNA concentrations and levels of cell-free mitochondrial DNA as in samples that had not been ultracentrifuged (*Figure 3G–I*). This indicates that the cell-free mitochondrial DNA recovered from cardiac organoids is not due to cell debris or mitochondrial fragments in the media.

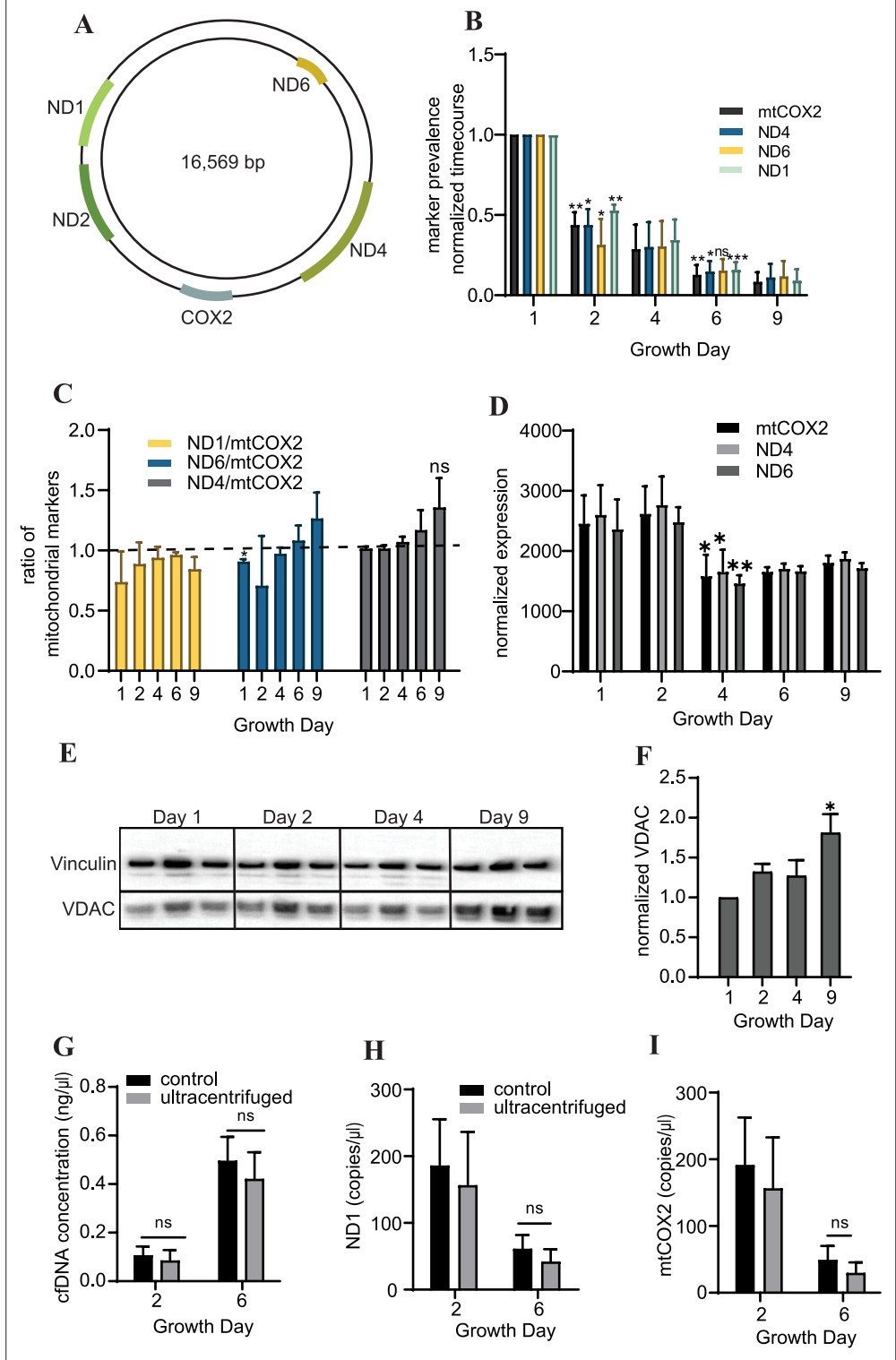

**Figure 3.** Cardiac organoids exhibit a time-dependent decrease in cell-free mitochondrial DNA abundance during growth. (**A**) Schematic showing the location of markers examined on the mitochondrial genome, adapted from Figure 1 of ***Uhler and Falkenberg, 2015***. (**B**) Relative abundance of mitochondrial markers ND1, ND4, ND6, and mtCOX2 in cardiac organoid-derived cfDNA, obtained using ddPCR. Shown are the average copies/μl of n = 3 biological replicates, each normalized to the values on day 1, to correct for batch variation. Samples on day 2 were compared to day 1 using a one-sample *t*-test comparing to a hypothetical value of 1.0. Day 6 samples were compared to day 2 values using an unpaired *t*-test with Welch's correction. *p<0.05; **p<0.01; ***p<0.001; ns,

*Figure 3 continued on next page*

*Figure 3 continued*

nonsignificant. (**C**) Ratio of the values obtained in panel (**B**) for each marker to mtCOX2. Significant deviations from a ratio of 1.0 indicating unequal marker abundance were determined using a one-sample *t*-test comparing to hypothetical value of 1.0. Shown are the averages of n = 3 biological replicates, + SD. (**D**) Expression of mitochondrial DNA markers at the tissue level in cardiac organoids during differentiation obtained using ddPCR, normalized to β-actin. (**E**) Western blot showing protein expression of the mitochondrial protein VDAC/Porin and loading control vinculin in n = 3 biological replicates of cardiac organoids taken at different timepoints during differentiation. (Vinculin loading control is the same for *Figure 4D*.) (**F**) Quantification of VDAC protein expression from the western blot in panel (**E**), normalized to vinculin. Samples were further normalized to day 1 and compared using a one-sample *t*-test with hypothetical value of 1.0 to test for significant deviation from day 1. (**G**) Concentration of cfDNA extracted from media conditioned by cardiac organoids on growth days 2 or 6 after ultracentrifugation (100,000 × *g* for 1 hr) compared to control (no ultracentrifugation). Abundance of mitochondrial markers ND1 (**H**) and mtCOX2 (**I**) in cfDNA derived from ultracentrifuged media conditioned by cardiac organoids during development, obtained using ddPCR. Graphs show average concentrations + SD. Samples were compared using an unpaired *t*-test with Welch's correction. *p<0.05; **p<0.01; ***p<0.001; ns, nonsignificant.

The online version of this article includes the following source data for figure 3:

**Source data 1.** Cardiac organoids exhibit a time-dependent decrease in cell-free mitochondrial DNA abundance during growth.

## Abundance of specific gDNA sequences is reflective of tissue-level changes in marker expression

Differentiating cardiac organoids show a switch-like increase in protein expression of the transcription factor Nkx2.5 between growth days 2 and 4 (*Figure 4A*). Intriguingly, we also observed an increase in cfDNA levels of Nkx2.5 during the same time frame using ddPCR (*Figure 4B*). We therefore wished to examine the prevalence of additional genomic regions within the cfDNA collected during cardiac organoid differentiation. The transcription factor p53, which is widely recognized for its role in cancer progression, has also been observed to regulate many components of cardiac transcription including Nkx2.5 (*Mak et al., 2017*). We observed that cfDNA levels of genomic p53 regions were increased later in cardiac differentiation (*Figure 4C*). However, in contrast to Nkx2.5, tissue-level expression of p53 decreased as cardiac organoids differentiated (*Figure 4D and E*). Although the endodermal marker Sox17 and stem cell marker Oct3/4 were both detectable by ddPCR in cardiac organoid-derived cfDNA, no significant changes with respect to cardiac differentiation were observed (*Figure 4—figure supplement 1*). Troponin T was not detectable in cardiac organoid-derived cfDNA. Together, these results suggest that specific regions of genomic DNA may be reflective of cardiac differentiation status.

## Doxorubicin causes severe cardiac organoid malformation in comparison to CPI-203

To determine how or whether mitochondrial and genomic cfDNA markers change in response to toxicity and chemical exposure, we treated cardiac organoids with the known cardiotoxicant doxorubicin (DOX) or CPI 203 (CPI) – an epigenetic bromodomain inhibitor with unclear cardiotoxic potential. The drugs were added to the culture medium on day 4 for 48 hr, and cfDNA was collected at two time points post drug addition (*Figure 5A*). By day 6, organoids treated with either of these drugs exhibited decreased organoid size and an increased ring of diffuse cells and debris compared to control tissues (*Figure 5B–D*). By day 9, the tissues treated with DOX exhibited severe toxicity and had nearly dissociated; whereas, CPI-treated tissues demonstrated tissue recovery with a nominal size decrease compared to controls (*Figure 5E–G*). By day 9, detectable bpm were significantly reduced in DOX-treated organoids in contrast to CPI-treated tissues where bpm were comparable to untreated tissues (*Figure 5H*). Both drugs resulted in decreased size of mature organoids, with DOX having a more severe effect than CPI (*Figure 5I*).

We wished next to know how these drug treatments impacted differentiation of the cardiac organoids. On day 6, levels of the cardiac transcription factor Nkx2.5 were substantially increased in samples treated with DOX (*Figure 5J*), while CPI-treated organoids did not significantly differ in expression compared to control. Together, these results indicate that treatment with DOX dysregulates cardiac gene expression and causes more severe toxicity in cardiac organoids than CPI in this model, at the

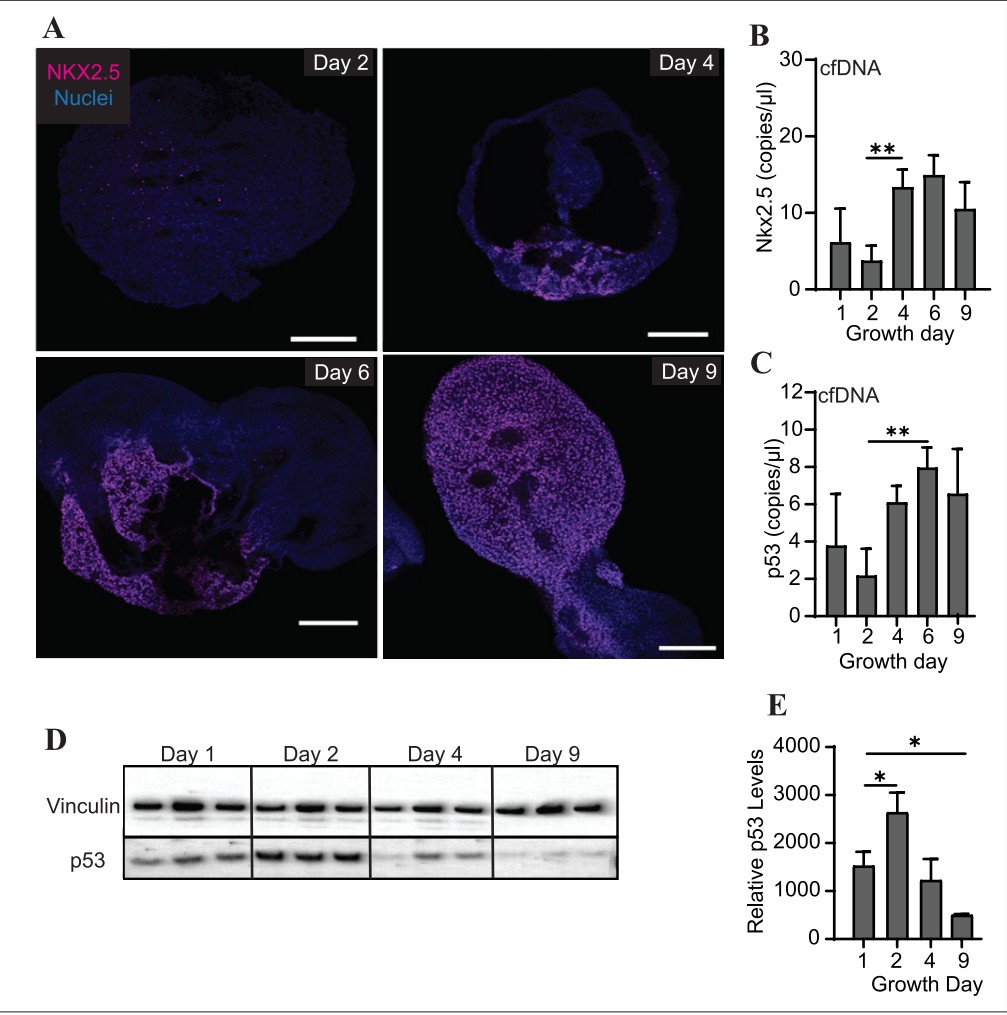

**Figure 4.** Abundance of specific gDNA sequences is reflective of tissue-level changes in marker expression. (**A**) Immunofluorescence staining of Nkx2.5 in cardiac organoids during differentiation. Scale bars represent 200 µm. (**B**) Nkx2.5 copies/µl in 1 ng of cardiac organoid-derived cfDNA at different time points during differentiation obtained using ddPCR, taken from n = 3 biological replicates. (**C**) p53 copies/µl in 1 ng of cardiac organoid-derived cfDNA at different time points during differentiation obtained using ddPCR, taken from n = 3 biological replicates. (**D**) Western blot showing p53 protein levels in cardiac organoids during differentiation. (Vinculin loading control is the same for *Figure 3E*.) (**E**) Quantification of the western blot in panel (**D**), normalized to Vinculin (n = 3 biological replicates). Graphs show average concentrations + SD. Samples were compared using an unpaired *t*-test with Welch's correction. *p<0.05; **p<0.01; ***p<0.001; ns, nonsignificant.

The online version of this article includes the following source data and figure supplement(s) for figure 4:

**Source data 1.** Copy number of Sox17 or Oct3/4 in cfDNA derived from cardiac organoids on different growth days during development.

**Figure supplement 1.** Abundance of specific gDNA sequences is reflective of tissue-level changes in marker expression.

**Figure supplement 1—source data 1.** Copy number of Sox17 or Oct3/4 in cfDNA derived from cardiac organoids on different growth days during development.

concentrations used. Notably, the timing of drug addition greatly impacted the degree of toxicity observed. Treating tissues with DOX later in development on day 7 resulted in less dramatic toxicity and no substantial changes in beating rate or tissue size compared to control. DOX still increased Nkx2.5 expression regardless of treatment time, while the impact of CPI on Nkx2.5 expression was less pronounced (*Figure 5—figure supplement 1*).

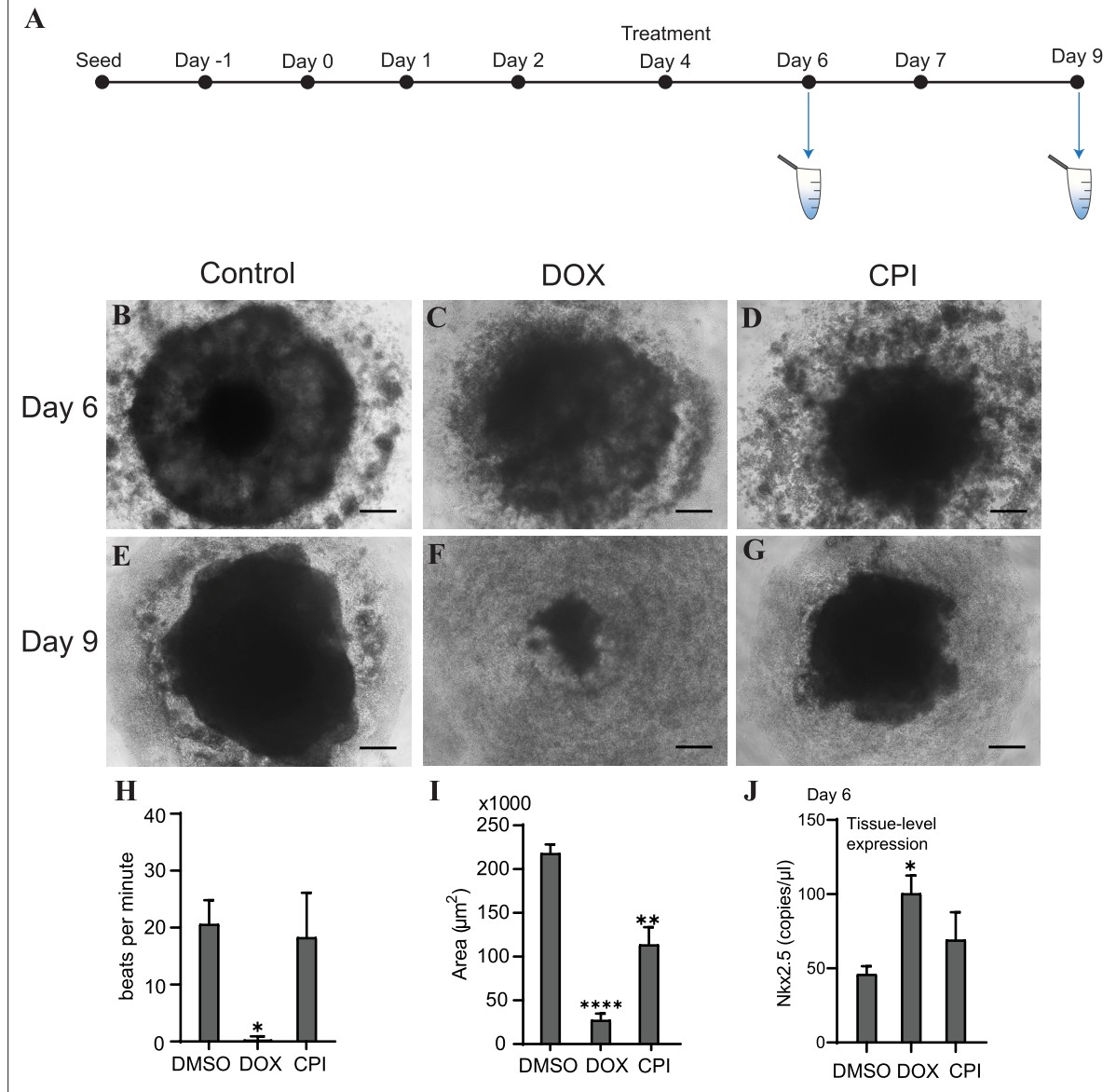

**Figure 5.** Doxorubicin (DOX) causes severe cardiac organoid malformation in comparison to CPI. (**A**) Schematic showing drug treatment and collection times of cfDNA during cardiac organoid growth. Representative images of tissues on day 6: control (**B**), DOX-treated (**C**), and CPI-treated (**D**). Representative images of tissues on day 9: control (**E**), DOX-treated (**F**), and CPI-treated (**G**). Scale bars represent 100 μm. (**H**) Beats per minute measured on growth day 9 for tissues treated with either DOX or CPI compared to DMSO-treated control. Shown are the average counts for 3–5 tissues each across three independent replicates. (**I**) Average tissue size of mature cardiac organoids treated with DOX or CPI measured on growth day 9. Shown are the average counts across three independent replicates (3–5 organoids per replicate). (**J**) ddPCR showing tissue-level expression of Nkx2.5 normalized to TBP on growth day 6 in cardiac organoids treated with DOX or CPI. Graphs show average concentrations + SD. Samples were compared using an unpaired *t*-test with Welch's correction. *p<0.05; **p<0.01; ***p<0.001; ns, nonsignificant. (Tissue morphology in response to treatment with DOX later in organoid growth is shown in *Figure 5—figure supplement 1*.)

The online version of this article includes the following source data and figure supplement(s) for figure 5:

**Source data 1.** Doxorubicin causes severe cardiac organoid malformation in comparison to CPI-203.

**Figure supplement 1.** Toxicity of DOX and CPI-203 in cardiac organoids is impacted by time of exposure.

**Figure supplement 1—source data 1.** Toxicity of DOX and CPI-203 in cardiac organoids is impacted by time of exposure.

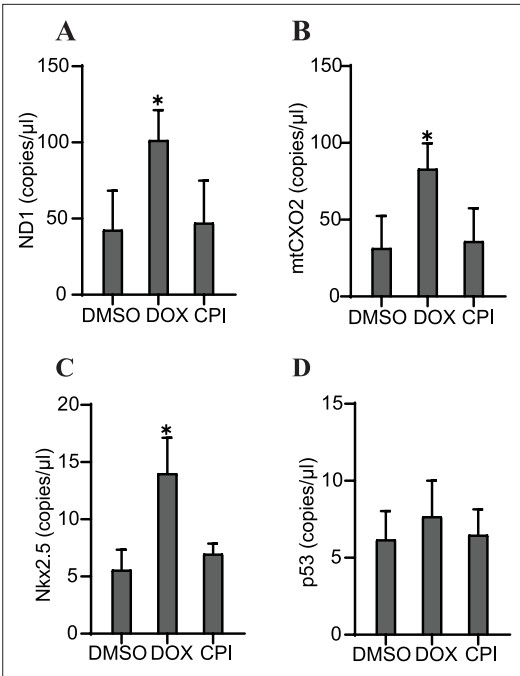

**Figure 6.** Specific cfDNA sequences may be predictive of toxicity in cardiac organoids. Abundance of ND1 (**A**), mtCOX2 (**B**), Nkx2.5 (**C**), or p53 (**D**) in 0.5 ng cfDNA collected from cardiac organoids on growth day 6 treated with doxorubicin (DOX) or CPI, obtained using ddPCR. Graphs show average concentrations + SD. Samples were compared using an unpaired *t*-test with Welch's correction. *p<0.05; **p<0.01; ***p<0.001; ns, nonsignificant. (Concentration of cfDNA and additional analysis of cfDNA sequences upon treatment with DOX later in growth are shown in *Figure 6—figure supplement 1*.)

The online version of this article includes the following source data and figure supplement(s) for figure 6:

**Source data 1.** Specific sequences of cfDNA may be predictive of toxicity in cardiac organoids.

**Figure supplement 1.** Concentration of cfDNA in response to early or late exposure to DOX or CPI-203.

**Figure supplement 1—source data 1.** Concentration of cfDNA in response to early or late exposure to DOX or CPI-203.

## Specific sequences of cfDNA may be predictive of toxicity in cardiac organoids

Our next step in the study investigation was to determine whether specific properties of cfDNA could be predictive of the outcomes we observed in each of these conditions: severe cardiac malformation (DOX), mild effects (CPI), or normal (control). Tissues that had been treated with DOX showed significantly increased levels of the mitochondrial markers ND1 and mtCOX2 on day 6 after 48 hr of drug exposure (*Figure 6A and B*). Nkx2.5 was also increased at the cfDNA level in DOX-treated samples (*Figure 6C*), while cfDNA levels of p53 were unchanged by toxicant addition (*Figure 6D*). The impact of DOX or CPI treatment on the abundance of these markers in cfDNA later in development was more subtle. In addition, the overall concentration of cfDNA was not significantly impacted by these drug treatments either early or late in development (*Figure 6—figure supplement 1*). Notably, the increases in ND1, mtCOX2, and Nkx2.5 in cfDNA collected from DOX-treated samples on day 6 preceded the detriment of the organoids we observed on day 9. Further, this increase was not observed in CPI-treated samples, which had largely recovered tissue morphology and beating function by day 9. Together, these results suggest that the abundance of specific markers (ND1, mtCOX2, and Nkx2.5) in cfDNA may be predictive of the severity of exposure outcomes in cardiac development.

## Discussion

Previously, cfDNA had been extracted successfully from pancreatic organoids derived from cancer patients (*Dantes et al., 2020*). However, whether cfDNA could be recovered from additional organoid types and the utility of these models for drug screening and predictive biomarker identification has been largely unexplored. Our study designed a method for rapid screening of cfDNA from cardiac organoids. We identified that decreases in cell-free mitochondrial DNA levels occur in response to cardiac differentiation. Further, we observed increases in specific cfDNA sequences corresponding to the genes Nkx2.5 and p53 upon cardiac organoid maturation. Our findings indicate that cfDNA concentration alone is not a reliable marker of toxicity. Rather, the prevalence of cell-free mitochondrial DNA and specific genomic regions may be more predictive of deleterious events. Specifically, cfDNA levels of ND1, mtCOX2, and Nkx2.5 in mature cardiac organoids may be potential predictive biomarkers of DOX-induced cardiotoxicity. We saw increases in the abundance of these markers on day 6 in DOX-treated samples, but not CPI-treated samples. By day 9, the organoids treated with DOX early in growth were severely malformed, while the CPI-treated organoids had mostly recovered.

Although our cardiac organoid model does not approach the complexity of cardiac growth in vivo, mapping which combinations of cfDNA markers reflect specific cardiotoxic outcomes would be a

valuable step towards implementing cfDNA as a clinical screening tool. In addition, organoid models permit the investigation of cfDNA in response to toxic compounds and chemicals with unclear bioactive potential, which could aid the development of cfDNA as a tool for detection of toxic exposures. Future work is needed to explore the impact of additional compounds and drugs on cardiac cfDNA, and what changes occur in other tissue types. Cell-free mitochondrial DNA output in response to differentiation and toxicity may be an important area of further investigation. The increase in cardiac organoid-derived cell-free mitochondrial DNA observed after treatment with DOX might reflect aberrant cardiomyocyte death. Accordingly, the decrease in cell-free mitochondrial DNA during cardiac organoid differentiation may represent preferential retention of cells with more mature mitochondria. Alternatively, the channels mPTP and VDAC may play a role in increased cfDNA release from stressed mitochondria (*Xian et al., 2022*). Fittingly, we observed that cell-free mitochondrial DNA markers (ND1, mtCOX2) were increased in organoids exposed to DOX. The exact mechanism of preferential cell-free mitochondrial DNA release by certain cell types is an intriguing area of future study.

In addition, identification of cfDNA sequences of nuclear origin is also critical for our understanding of the cell-free genome. Differential abundance of cell-free genomic DNA may reflect regions of DNA protected from DNases by transcription factor binding or epigenetic modifications (*Ulz et al., 2019*). The transcription factor p53, which is widely recognized for its role in cancer progression, also has demonstrated involvement in cardiac tissues and interacts with Nkx2.5 (*Kojic et al., 2015*). Although both of these markers increased at the cfDNA level as the organoids matured, we observed with interest that the overall expression of Nkx2.5 increased whereas that of p53 decreased at the tissue level during maturation. Analysis and identification of multiple combinations of cfDNA markers may ultimately be necessary to effectively distinguish events at the tissue level. The cfDNA we recovered from cardiac organoids consisted of a heterogeneous distribution of fragment lengths that ranged from approximately 200–6000 base pairs. The shorter fragments are potentially indicative of DNA that has been cleaved during apoptosis, while the longer lengths could result from necrotic processes (*Aucamp et al., 2018*). However, cfDNA can also be expelled actively from healthy cells in the form of extracellular vesicles or nucleoprotein complexes. Further, different DNases may contribute uniquely to patterns of fragmentation (*Han et al., 2020*). Tracing individual fragments back to the cellular event triggering their release would be a valuable area of future research.

In the bloodstream, multiple cell types contribute to the cfDNA population, with a large fraction released by white blood cells and erythrocyte progenitor cells (*Moss et al., 2018*). This complexity highlights the need to identify key cfDNA sequences that can be used as clinical markers for specific conditions. Although media conditioned by organoids *in culture* does not approach the complexity of human blood or bodily fluids, organoids provide a tractable system for identifying potential cfDNA targets in response to known tissue-level events. These cfDNA markers may be valuable targets for further screening in clinical samples. Although cfDNA concentration is known to be elevated in cardiac disease patients (*Polina et al., 2020*), we did not observe a significant increase in overall cfDNA concentration in response to phenotypic changes. Although cell death is one route of cfDNA release, alternative mechanisms exist through which cfDNA concentration may increase in the absence of cell death, such as association with extracellular vesicles (*Kustanovich et al., 2019*; *Aucamp et al., 2018*). In clinical samples, cfDNA concentration can fluctuate substantially between individuals and increase in response to a number of conditions including cancer (*Salvi et al., 2016*), exercise (*Breitbach et al., 2012*), aging, and recent surgery (*Aucamp et al., 2018*). These observations combined with our results indicate that cfDNA concentration alone may not be an optimal parameter for targeted disease diagnosis. Rather, screening for specific cfDNA sequences may be more indicative of certain conditions.

In this study, we have demonstrated that cfDNA can be reliably extracted from cardiac organoid models. Further, we present the tractability of organoid systems for screening cfDNA markers in response to tissue-level events, including differentiation and malformation. This workflow has the potential for rapid screening of cfDNA from toxicity and disease models and could likely be extended to additional organoid systems. Our future efforts include the identification of additional cfDNA sequences that are impacted by differentiation and understanding how different chemicals impact the cfDNA profile of mature and developing cardiac tissues. Ultimately, our approach could contribute to establishing a database of cfDNA markers that map to specific disease processes or toxicants, which would hold promise for early, non-invasive detection of disease.

## Study limitations and future directions

Here, cfDNA samples were collected at a small number of timepoints along the development of the organoids. Future research might explore more frequent samplings of cfDNA to obtain a better understanding of how cfDNA changes in time during organoid growth. In addition, it is unclear to what degree cfDNA may be degraded between the time it is released from the tissues to the time of collection. Further, although we attempted to minimize DNA degradation post-collection by limiting freeze–thaw cycles and using good sample storage practices, we cannot rule out that longer fragments of DNA could have broken down into shorter ones over time. In addition, although the Maxwell ccfDNA automated extraction kit is one of the better tools currently available for cfDNA extraction (*Sorber et al., 2017*), we cannot rule out potential bias towards short fragments. The electropherograms which showed cfDNA size distribution are qualitative and cannot be taken as a definitive statement of fragment profiles, especially since events such as post-collection degradation could be missed through this analysis. Future studies might investigate more closely the molecular dynamics of cfDNA release and the timing of its degradation, and also how degradation rate may vary in culture media versus the bloodstream.

To account for the growing size of the organoids, we used total tissue-level genomic DNA concentration to normalize cfDNA concentrations as the dense nature of the tissues hindered their dissociation and single-cell counts via enzymatic or mechanical means. However, this is only an approximation and may have introduced errors resulting from multinucleated cells, aneuploidy, or cells undergoing division. Future work might seek to optimize methods of normalizing cfDNA concentration to further improve our knowledge of how cfDNA concentration is impacted by tissue-level events. Unlike tissue-level analyses which examine expression of RNA transcripts converted to complementary DNA (cDNA), cell-free DNA (cfDNA) consists of expelled DNA fragments that are present in the extracellular media. It is currently unclear whether certain fragment sequences might be present more consistently than others, so it is therefore difficult to normalize ddPCR analyses of cfDNA copy number to a housekeeping gene. This is a possible source of error in determining the abundance of specific cfDNA sequences. Future studies of normalization and copy number analyses in free nucleic acids would be valuable towards precise quantitative assessments of cfDNA sequences.

In this study, only a handful of cfDNA sequences were assessed. In addition, our study only assessed organoids derived from one cell type (H9 ESCs) in a limited number of batches (n = 3). However, we feel this work has demonstrated that cfDNA can be successfully recovered from cardiac organoids in quantities sufficient for characterization and quantification. This lays the groundwork for further studies of cfDNA from additional organoid systems in response to toxicant treatments, as well as broader global analyses such as microarrays or sequencing.

# Materials and methods

## Cell culture

WA09 (H9) human embryonic stem cell line was purchased from WiCell. Authentication of cell lines and mycoplasma was conducted by WiCell. H9 human embryonic stem cells (hESCs) were maintained in mTeSR+ medium (Stemcell Technologies, 100-0276) at 37°C and 5% $CO_2$. Cells were passaged between 60 and 80% confluence using 0.5 mM EDTA (Gibco, 15575) onto plates pre-coated with growth factor-reduced Matrigel (Corning, 354230) for 30 min at 37°C in DMEM/F:12 (Gibco, 11330-032) at a concentration of 1.2 µL/mL $cm^2$. Cultures were passaged a maximum of 10 times provided no morphological changes indicative of differentiation were present.

## Cardiac organoid generation

Cardiac organoids were generated using a previously established protocol (*Israeli et al., 2020*). Briefly, H9 hESCs were detached using TrypLE (Gibco, A12177-01) and resuspended at a density of 100,000 cells/mL in E8 media (Gibco, A15169-01) + 10 µM Y27632 (Tocris, 129830-38-2). 100 µL of cell suspension was added per well to a 96-well round-bottom ultra-low attachment plate (Thermo Fisher, 174925) and centrifuged for 5 min at 200 × *g* to coalesce the cells. The day of seeding was defined as day 2 of growth. On day 1, the media was changed by removing 50 µL media from each well and adding 200 µL fresh prewarmed E8 media. On day 0, 166 µL media was removed and replaced with 166 µL RPMI 1640 (Gibco, 11875-093) supplemented with 1x B27 (no insulin; Gibco, A18956-01),

CHIR 99021 (4 µM; Selleck, S2924), BMP4 (0.36 pM; BioTechne, 314BP-010/CF), and ActA (0.08 pM; BioTechne, 338-AC-010/CF). On day 1, 166 µL of media was removed and replaced with 166 µL RPMI 1640 supplemented with B27 (no insulin). On day 2, 166 µL of media was removed and replaced with 166 µL RPMI 1640 supplemented with B27 (no insulin) and IWR-1-endo (5 µM; Selleck, S7086), and incubated at 37°C for 48 hr. On day 6, 166 µL of media was removed and replaced with 166 µL of RPMI 1640 supplemented with 1x B27 (with insulin; Gibco, 17504-044). On day 7, 166 µL of media was removed and replaced with 166 µL RPMI 1640 supplemented with B27 (with insulin) and 4 µM CHIR99021, and incubated for 1 hr at 37°C, then the media was again replaced as described above using RPMI 1640 supplemented with B27 (with insulin; no CHIR). The plate was incubated at 37°C for 48 hr, and the cardiac organoids were considered mature on day 9. To examine the effects of chemical addition, doxorubicin (DOX, 1 nM; Sigma, D1515) or CPI-203 (0.5 µM; Tocris, 5331) were applied for 48 hr on either day 4 or 7 of cardiac organoid development (control, 1:1000 DMSO) in culture media.

## Nucleic acid extraction

RNA and gDNA from organoids (Promega, miRNA Tissue Kit AS1460; Tissue DNA Kit AS1610) and cfDNA from media (Promega, ccfDNA Plasma Kit AS1480) were extracted using the Promega Maxwell RSC instrument. RNA quality and purity were assessed using a NanoDrop 2000 spectrophotometer, and DNA concentrations were measured using the Promega QuantiFluor dsDNA System. A minimum of 3–5 organoids were used per biological replicate for RNA and gDNA extractions. RNA was converted to cDNA using the iScript Reverse Transcription Kit (Bio-Rad). CfDNA from cardiac organoids was collected from media conditioned by 10 tissues (2–3 technical replicates per batch, 3 independent batches). Prior to cfDNA extraction, all conditioned media was centrifuged for 10 min at $1600 \times g$, the top 800 µL supernatant centrifuged 10 min at $16,000 \times g$, and the top 400 µL collected and stored at –80°C until extraction. Prior to extraction, conditioned media was thawed at 4°C to prevent degradation.

## Quantification of cfDNA concentration, fragment analysis, and gene expression

Nucleic acid concentration was measured using the QuantiFluor ONE dsDNA System (Promega, E4871). Fragment analysis of cfDNA size was performed using the 5200 Fragment Analyzer System, Agilent Inc using the HS Large Fragment 50 kb Kit (Agilent, DNF-464-0500). Gene expression and cfDNA copy number were evaluated using droplet digital PCR (ddPCR; HEX/FAM system) using 0.5 ng or 1 ng of cDNA or cfDNA per reaction. Prior to quantification, cfDNA concentrations were diluted to 0.05 ng/µL and 10 µL added per reaction. ddPCR was performed using 25 µL reaction volumes with ddPCR Supermix for Probes (no dUTP; Bio-Rad) and 1 µL each of the following probes: Troponin T (dHsaCPE5052345, Bio-Rad), Nkx2.5 (dHsaCPE5042098, Bio-Rad), Oct 3/4 (POU5F1, dHsaCPE5191335, Bio-Rad), ND1 (dHsaCPE5029121, Bio-Rad), MT-COX2 (dHsaCPE5192286), p53 (TP53, dHsaCPE5037521, Bio-Rad), SOX17 (dHsaCPE5039713, Bio-Rad), ND6 (dHsaCNS941916401, Bio-Rad), ND4 (dHsaCNS186386931, Bio-Rad), GAPDH (dHsaCPE5031597, Bio-Rad), TPB (dHsaCPE5058363, Bio-Rad), and β-actin (dHsaCPE5190200, Bio-Rad). Droplets were generated using the Automated Droplet Generator (Bio-Rad, CA) and PCR performed using annealing conditions of 60°C for 1 min, 40 cycles. Droplets were read using the QX200 Droplet Reader (Bio-Rad). Thresholding was applied equally across all sample conditions under comparison for each probe.

## Protein detection and quantification

A minimum of five organoids per batch were rinsed 1× with PBS and homogenized in Pierce RIPA (Thermo Scientific, 89900) supplemented with 25× cOmplete Protease Inhibitor, EDTA-free (Roche, 11836170001) and centrifuged for 15 min at $14,000 \times g$ at 4 °C. The supernatant was removed and stored at –80°C until analysis. Protein concentration was calculated using the Pierce BCA Protein Assay (Thermo Scientific, 23225). 3.5 µg protein was incubated for 10 min at 70°C with NuPAGE LDS Sample Buffer (Invitrogen, NP0007) and NuPAGE Sample Reducing Agent (Invitrogen, NP 0009), loaded onto a NuPAGE 4 to 12%, Bis-Tris gel (Invitrogen, NP0336BOX), and run for 35 min at 200 V in 1× NuPAGE MES Run Buffer (Invitrogen, NP0002) plus NuPAGE Antioxidant (Invitrogen, NP0005). The proteins were transferred to nitrocellulose membranes using the iBlot Gel Transfer Device (Thermo Fisher). Protein transfer was visualized using Ponceau S solution (Sigma, P7170), and the membranes were

cut into regions containing the proteins of interest. Membranes were blocked for 15 min in EveryBlot Blocking Reagent (Bio-Rad, 120110020) and incubated with primary antibody overnight at 4°C diluted 1:1000 in EveryBlot Blocking Reagent unless otherwise specified: rabbit anti-vinculin (Novus, NBP2-20859), rabbit anti-Nkx2.5 (Cell Signaling, 8792T), rabbit anti-VDAC (Abcam, ab15895), mouse anti-p53 (1:500; Novus, NBP2-29453), rabbit anti-MEF2C (Cell Signaling, 5030), rabbit anti-GATA-6 (Cell Signaling, 5851), rabbit anti-α-actinin (Cell Signaling, 6487), rabbit anti-GAPDH (Bio-Rad, VPA00187). Following primary antibody incubation, membranes were washed 3 × 5 min in 1× Tris-buffered saline (TBS; Bio-Rad, 1706435) plus 0.1% tween-20 (Sigma, P7949) (TBST) and incubated for 45 min at room temperature with secondary antibody diluted 1:5000 in EveryBlot Blocking Reagent (Bio-Rad): goat-anti-rabbit HRP (Novus, NB7160) or goat-anti-mouse HRP (Invitrogen, 32230). Membranes were washed 3 × 15 min in TBST and HRP visualized using SuperSignal West Pico Plus Chemiluminescent Substrate (Thermo Scientific, 34580). Bands were quantified in ImageJ (*Schneider et al., 2012*) using a rolling ball background subtraction with $r = 50$, followed by normalization to vinculin.

## Immunostaining

We used a previously described protocol (*Matsumoto et al., 2019*) for immunostaining and clearing of the organoids. Briefly, the organoids were fixed overnight in 4% paraformaldehyde followed by delipidation by successive overnight incubations in CUBIC-L solution (50%, 100%) at 37°C. The organoids were then blocked in 3% bovine serum albumin and incubated for 2 d in rabbit anti-Nkx2.5 primary antibody diluted 1:200 in blocking solution. Excess primary antibody was removed using PBST (0.2% Triton X-100 diluted in 1× PBS). The organoids were then incubated again for 2 d in secondary antibody (1:200, Alexa 647 goat-anti-rabbit; Invitrogen, A21244) and DAPI (1 µg/mL; Sigma, P9542) diluted in blocking solution. Samples were cleared for imaging in 96-well optical plates (Thermo Fisher, M33089) using CUBIC-R solution as described previously (*Matsumoto et al., 2019*). Confocal images were obtained on a Zeiss LSM 880 inverted confocal microscope with AiryScan (Jena, Germany).

## Statistical analysis

Data were analyzed using GraphPad Prism version 9.0.0 for Windows, GraphPad Software, San Diego, CA (https://www.graphpad.com/). Averages of sample conditions were compared using an unpaired *t*-test with Welch's correction. *$p<0.05$; **$p<0.01$; ***$p<0.001$; ns, nonsignificant. In instances where data were normalized to the initial timepoint (day 1) to account for batch variation, or ratios were compared, a one-sample *t*-test was used to compare the average of each normalized condition to a hypothetical value of 1.0, to detect significant deviation from the day 1 starting point. *$p<0.05$; **$p<0.01$; ***$p<0.001$; ns, nonsignificant. All graphs show average + SD unless otherwise stated. Three independent biological replicates (separate batches of organoids cultured consecutively on different days) were performed for each experiment.

## Acknowledgements

We thank the NIEHS Molecular Genomics Core facility, and in particular Wesley Gladwell, for generous assistance with the analysis and setup of Droplet Digital PCR experiments. Additionally, we are very thankful to Dr. Xian Wu for training in cardiac organoid culture. Also, we express our appreciation for the NIEHS Fluorescence Microscopy and Imaging Center for their assistance with fluorescence imaging of the organoids. In addition, we wish to acknowledge and thank Dr. Ian Chen for training and assistance in immunofluorescence staining and tissue-clearing protocols.

# Additional information

### Funding

| Funder | Grant reference number | Author |
| --- | --- | --- |
| National Institutes of Health | ES103378-01 | Erik Tokar |

| Funder | Grant reference number | Author |
|---|---|---|
| National Institutes of Health | ES102546 | Kevin Gerrish |

The funders had no role in study design, data collection and interpretation, or the decision to submit the work for publication.

## Author contributions

Brian Silver, Conceptualization, Data curation, Formal analysis, Investigation, Methodology, Writing - original draft, Writing - review and editing; Kevin Gerrish, Erik Tokar, Conceptualization, Formal analysis, Supervision, Funding acquisition, Writing - original draft, Writing - review and editing

## Author ORCIDs

Brian Silver ⓘ http://orcid.org/0000-0002-4480-5009
Kevin Gerrish ⓘ http://orcid.org/0000-0002-4236-5329
Erik Tokar ⓘ http://orcid.org/0000-0002-1668-2830

## Decision letter and Author response

Decision letter https://doi.org/10.7554/eLife.83532.sa1
Author response https://doi.org/10.7554/eLife.83532.sa2

---

# Additional files

## Supplementary files

• MDAR checklist

## Data availability

All data generated or analysed during this study are included in the manuscript and supporting file.

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
