## [Editor Report]

This important study presents a comprehensive investigation into cell-free DNA (cfDNA) within the context of 3D cell cultures, offering valuable insights into this emerging research area. Despite being an exploratory study, the findings provide compelling evidence that continued progress holds the potential for the establishment of versatile 3D cell culture cfDNA assays. Such assays could serve as invaluable research and clinical tools, enabling monitoring of organoid growth and development, enhanced characterization of tissue dynamics, and ultimately facilitating the identification of novel biomarkers.

---

## [Decision Letter]

**Decision letter after peer review:**

Thank you for submitting your article "Cell-free DNA as a potential biomarker of differentiation and toxicity in cardiac organoids" for consideration by *eLife*. Your article has been reviewed by 3 peer reviewers, including Abel Bronkhorst as the Reviewing Editor and Reviewer #1, and the evaluation has been overseen by Didier Stainier as the Senior Editor.

Essential revisions:

1) Please provide a point-by-point response to all reviewer comments.

2) Note that additional experiments have been requested by the reviewers. Please perform these experiments or provide a convincing rebuttal.

*Reviewer #1:*

CfDNA molecules possess various physicochemical features that correlate with the cellular origin and a wide range of disease indications, making it an ideal biomarker. However, systematic mapping of the information available through cfDNA characterization is complicated by the immense complexity of in vivo systems. While cell culture models represent only a limited view into complex biological systems, they have still proven to be extremely useful in many cases and a growing number of studies indicate their utility in cfDNA analysis.

in vitro research on cfDNA has to date focused mainly on 2D cell cultures. Therefore, this work by Silver et al. is among only a small number of studies that profiled cfDNA in 3D cell cultures and represents one of the most detailed assessments to date. They demonstrated that (i) cfDNA release correlates with differentiation; (ii) cell-free mitochondrial DNA levels correlate with the development of organoids; (iii) the sequence composition of cfDNA fluctuates during growth; and (iv) specific cfDNA features correlates with toxic drug treatment.

Based on these interesting observations the authors conclude that cfDNA profiling may not only serve as a useful tool to monitor the growth and development of organoids but may be leveraged to gain insight into tissue dynamics and potentially identify new biomarkers. The authors acknowledge that their study is only an explorative study and that more research is needed before in vitro data will reveal accurate and practically useful information about the in vivo setting. However, the various issues that may confound the results obtained in their study, and the general limitations of 3D cell models in relation to cfDNA analysis, are not clearly outlined. For example, (i) factors that affect the normalization of cfDNA concentration measurements, (ii) the influence of "natural" cell death on the measurement of drug-induced cell death-related cfDNA features, (iii) the effects of DNA degradation over time, (iv) important dynamic changes in cfDNA properties that are missed due to widely spaced measurement time-points.

1. Page 3, lines 53-54: "Several measures including concentration, fragment size, and sequence can be used to characterize cfDNA, and imply information about the cells from which it originated [1]."

• Several epigenetic features of cfDNA (e.g. fragment end-point motifs, nucleosome spacing, topological features, differentially methylated regions, and histone modifications) enable enhanced cell-of-origin identification. These features are increasingly characterized and could be mentioned here.

2. Page 3, lines 55-57: "Still, the current diagnostic uses of cfDNA are primarily limited to specific applications such as the detection of fetal chromosome abnormalities responsible for Down syndrome [6] and fetal gender determination from maternal blood [7]."

• There are a number of FDA-approved cancer cfDNA assays for routine clinical use. Most of these are companion tests but could be mentioned here, as the clinical use of cfDNA assays is not primarily limited to non-invasive prenatal testing (NIPT).

3. Page 5, lines 98-101: "To account for changes in cell number during organoid growth, we normalized the concentrations of cfDNA to levels of genomic DNA (gDNA) collected on the same growth days. Normalized cfDNA was highest early in development (Figure 2C), indicating that cfDNA output may be higher in tissues comprised of less differentiated cells."

Please expand this section after considering the following points:

• There is no data linking the timing of specific cellular events with the extracellular occurrence of DNA fragments. For example, cell division, differentiation, apoptosis, necrosis, active release, drug-induced death, etc., may be initiated at one point in time, while DNA may only be released into the growth medium at a much later time point. Similarly, a major portion of specific DNA fragments released due to specific reasons may already be degraded at the time of measurement. How is this factored into cfDNA measurements and normalization of measurements?

• DNA content does not necessarily reflect cell number, for example during specific cell division phases or aneuploidy

• Is it possible and would it be better to perform both gDNA analysis and total cell counts?

• Would it be better to perform more frequent cfDNA assessments, e.g., every couple of hours?

4. Page 5, lines 102-103: "The recovered cfDNA consisted of a broad distribution of fragments ranging from approximately 200 to 6000 base pairs (Figure 2D)."

• Some comments here:

• The cfDNA size profile is interesting. The short fragments demonstrate an apoptotic laddering pattern, but there is also a population of long cfDNA fragments, which may indicate an origin from apoptosis. Could the authors elaborate on this?

• A recent publication on the characterization of cfDNA in 2D cell cultures suggested that shorter fragments and longer fragments may originate from the same process and that longer fragments are degraded into shorter fragments over time.

o This paper also shows that the apparently high concentration of longer cfDNA fragments as seen in electropherograms may be an artefact of the method. Could the authors comment on this?

• If it is true that longer fragments also contain important biological information, this information could be missed in some cases, for example:

o The use of extraction kits that are biased toward the capture of short fragments

o DNA sequencing techniques that are limited to short reads

o cfDNA sizing using PCR techniques

• The method used for cfDNA sizing is not discussed in the methods and materials section.

• It would have been interesting to not only see the cfDNA size profile on one day but how it changes over the course of incubation. Perhaps the authors can speculate on what insights this may reveal.

5. Another factor that may complicate the characterization of organoid-derived cfDNA:

• Literature suggests that natural levels of apoptosis /necrosis increase over the course of organoid growth and development. How does one differentiate between cfDNA fragments originating from "natural" vs "induced" cell death? Is this a confounding factor for analysis?

*Reviewer #2:*

Silver et al. demonstrated that cell-free DNA can be detected from cultures of cardiac organoids. This subject has not been explored and presents a real interest towards the possible use of organoids to detect markers according to the physiological or pathological evolution of the original tissues. So they tried to assess the possibility of cardiotoxicity of a drug from organoid cultures. The writing of the manuscript is of good quality, concise, and the presentation of the results is clear. However, the manuscript has many flaws including many inaccuracies; and authors should address the following major concerns:

1. The analysis of free mitochondrial DNA raises many questions: (i), there is no technical explanation for standardization, calibration, and direct comparison of the quantification of the number of copies of different regions of the mitochondrial genome. The quantification of mitochondrial DNA by the quantitative PCR technique must be determined with caution; (ii) the readers may remain confused regarding the method used to determine DNA concentration; (iii) the reviewers is confused throughout the manuscript in respect to characterize gene expression (RT-PCR) and cfDNA amount (qPCR); (iv), the authors state in the abstract that there is a decrease in the fraction of free mitochondrial DNA during development organoids which is not proven by the results; (v), on what basis can the authors say that this identifies a unique signature of cardiac differentiation if they did not make a comparison with other cell types; (vi), the authors do not comment on why the expression of the four mitochondrial regions is equivalent, which is counterintuitive; (vii), Figure 3C shows standard variations when it looks impossible when showing ratios; and (viii), there is no discussion when comparing amount of cfDNA from both origins while they showed an apparent similar level, especially in regards to the literature comparing for instance cfDNA from mitochondria or nucleus.

2. How is it possible that wild-type DNA, in this case P53, can show a difference in quantity by quantitative PCR analysis during differentiation?

3. No or very few indications specified the number of passages or the number of organoids within the same culture. This is crucial and must be filled in.

*Reviewer #3:*

The article analyses for the first time the kinetics of different cfDNA species during the development of cardiac organoids generated from H9 human embryogenic stem cells. The stimulation of the cells with Doxorubicin or CPI, as a model of toxic exposure, is a relevant approach to identifying the validity of cfDNA as biomarkers of toxic events. The authors show an increase in cfDNA and a decrease in mtDNA during cardiac organoid differentiation. However, the research relies on a single cell line, and the number of biological replicates with n = 3 is very small. The inclusion of additional two or three replicates is required to validate the finding. Moreover, the inclusion of another selected cell line would strengthen the finding.

The article is well-written and the authors used various methodological approaches to prove their findings. However, with respect to the main message, namely, the increase of specific cfDNA targets in response to DOX treatment, the reviewer has some concerns. Those are related to statistical aspects as well as methodological as described below.

– In figures 1-4 the authors compare paired samples over time. However, in their figure legends, the authors indicate that an unpaired test was used. Does this have any rationale like missing samples? The single point of the samples should be included in the figure.

– In all figure legends it is noted that t-tests with Welch´s correction were used unless otherwise indicated. The authors could indicate if a t-test (homogenous variances between samples) or a welch´s t-test (not normal variances) was used.

– The authors should additionally highlight which of the results are related to cfDNA or cDNA analysis. For example, Figures 4 B and C relate to cfDNA (labelled with cfDNA). Figure 5 J relates to cDNA (not labelled). Figure 6 (not labelled with cfDNA).

– Figure 3 B, please clarify if the relative abundance of mtDNA relates to total cfDNA or one of the reference genes.

– Figure 4 should be restructured. Figure 4B, C, F, and G belong together, and D and E belong together. As long as F and G display cfDNA analysis (should be highlighted). Aftewards the legend text could be reduced relevantly.

– Figure 6: The copy number of the different cfDNA targets relies relatively on the input material (0.5 or 1 ng were included in the ddPCR assay). Especially for Figure 6 and the discussion that ND1 and NKx2.5 increase in response to DOX treatment, but not the total amount of cfDNA and not p53. Can this result be strengthened by the analysis of GAPDH or TBP in the same sample?

– The reviewer feels that 3 more replicates are needed to validate the finding of different cfDNA release in response to DOX. Moreover, a comparison to another suitable and well-selected cell line would be welcome.

---

## [Author Response]

Reviewer #1:1. Page 3, lines 53-54: "Several measures including concentration, fragment size, and sequence can be used to characterize cfDNA, and imply information about the cells from which it originated [1]."• Several epigenetic features of cfDNA (e.g. fragment end-point motifs, nucleosome spacing, topological features, differentially methylated regions, and histone modifications) enable enhanced cell-of-origin identification. These features are increasingly characterized and could be mentioned here.

The reviewer raises an excellent point. Epigenetic modifications are indeed an important area of cfDNA research in addition to the characterizations we mentioned. We have added this information and the additional supporting references as shown below.

“Several measures including concentration, fragment size, sequence, and epigenetic modifications such as differential methylation, histone modification, and nucleosome spacing can be used to characterize cfDNA,and imply information about the cells from which it originated [1, 6, 7].”

2. Page 3, lines 55-57: "Still, the current diagnostic uses of cfDNA are primarily limited to specific applications such as the detection of fetal chromosome abnormalities responsible for Down syndrome [6] and fetal gender determination from maternal blood [7]."• There are a number of FDA-approved cancer cfDNA assays for routine clinical use. Most of these are companion tests but could be mentioned here, as the clinical use of cfDNA assays is not primarily limited to non-invasive prenatal testing (NIPT).

We thank the reviewer for raising this point and have revised and expanded this section to include the additional mentioned assays for cancer. In addition, we have reworded these statements to more accurately reflect our view that cfDNA likely has many additional uses in noninvasive testing beyond that which is currently available.

“Currently, cfDNA already has some diagnostic uses including noninvasive prenatal testing for chromosome abnormalities [8], fetal gender determination [9], and cfDNA-based tests for the detection of mutations and cancer screening [10, 11]. Although the clinical utility and accuracy of these tests in the prediction of cancer is uncertain in some cases [12], clinical trials are promising and some FDA-approved tests are already available [10]. Still, we have likely yet to unlock the full spectrum of clinical usage for cfDNA in noninvasive diagnostics.”

3. Page 5, lines 98-101: "To account for changes in cell number during organoid growth, we normalized the concentrations of cfDNA to levels of genomic DNA (gDNA) collected on the same growth days. Normalized cfDNA was highest early in development (Figure 2C), indicating that cfDNA output may be higher in tissues comprised of less differentiated cells."Please expand this section after considering the following points:• There is no data linking the timing of specific cellular events with the extracellular occurrence of DNA fragments. For example, cell division, differentiation, apoptosis, necrosis, active release, drug-induced death, etc., may be initiated at one point in time, while DNA may only be released into the growth medium at a much later time point.

We thank the reviewers for bringing up this point, and we agree that our data do not necessarily link cfDNA abundance to cellular differentiation status or specific events, since multiple cell types are present and change dynamically throughout the cardiac organoid development process. Accordingly, we have revised the statement in lines 98-101 to clarify that our cfDNA observations are associated with organoid growth day, and not necessarily specific cellular events. Although an assessment of the precise cellular origin of cfDNA and the impact of specific differentiation events on cfDNA is beyond the scope of this study, we feel additional future research in this area would be of great value. To further address the points discussed by the reviewer, we have added a limitations section in our discussion (page 13) to further clarify the boundaries of our current study and suggest future research topics.

Lines 98-101: “Normalized cfDNA was highest early in development (**Figure 2C**), indicating that cfDNA output may be higher in tissues on growth day 1, which are comprised of less differentiated cells.”

Similarly, a major portion of specific DNA fragments released due to specific reasons may already be degraded at the time of measurement. How is this factored into cfDNA measurements and normalization of measurements?

The reviewer raises a thoughtful point. Although we attempted to minimize potential degradation by limiting freeze/thaw cycles and keeping conditioned media and cfDNA at 4^o^C between steps, a molecular description of how cfDNA is degraded between the time of release and collection was not studied here. We feel this would be beyond the scope of our current study as we were more interested in the validation of cardiac organoids as a tool for identifying potential candidate cfDNA biomarkers. Future research might explore the molecular dynamics of cfDNA release and degradation, and how degradation rate may vary in culture media vs the bloodstream. To highlight this important area of future research, we have included the following statements in the limitations section of our discussion.

“… it is unclear to what degree cfDNA may be degraded between the time it is released from the tissues to the time of collection. Future studies might investigate more closely the molecular dynamics of cfDNA release and the timing of its degradation, and also how degradation rate may vary in culture media versus the bloodstream.”

• DNA content does not necessarily reflect cell number, for example during specific cell division phases or aneuploidy• Is it possible and would it be better to perform both gDNA analysis and total cell counts?

We thank the reviewer for this point. Accounting for cell number using gDNA may indeed have slight inaccuracies due to events such as the reviewer mentions. We previously attempted single cell counts but were unable to dissociate the cells in the dense organoids via mechanical and enzymatic means without damaging them, which hindered obtaining accurate single cell counts. We felt using gDNA content was our best attempt to normalize cfDNA to account for an estimate of growing tissue size. To clarify that this could be a potential source of error, we have included the following statement in the limitations section of our discussion:

“To account for the growing size of the organoids, we used total tissue-level genomic DNA concentration to normalize cfDNA concentrations, as the dense nature of the tissues hindered their dissociation and single-cell counts via enzymatic or mechanical means. However, this is only an approximation, and may have introduced errors resulting from multinucleated cells, aneuploidy, or cells undergoing division. Future work might seek to optimize methods of normalizing cfDNA concentration to further improve our knowledge of how cfDNA concentration is impacted by tissue-level events.”

• Would it be better to perform more frequent cfDNA assessments, e.g., every couple of hours?

Although an interesting question, current cardiac organoid culture methods require precise timing of media changes (24 or 48 hr). It is unclear whether organoids can be successfully generated with more frequent media collections. Additionally, because the organoids are cultured over a 12-day period, taking more frequent timepoints (every couple of hours) would yield a very large number of samples that would exceed our current capabilities in terms of cost and time. Although beyond the scope of our current study, we feel the experiment the reviewer suggests would indeed be a valuable area of future research. To highlight this point, we have added the following statements to our discussion:

“Here, cfDNA samples were collected at a small number of timepoints along the development of the organoids. Future research might explore more frequent samplings of cfDNA, to obtain a better understanding of how cfDNA changes in time during organoid growth.”

4. Page 5, lines 102-103: "The recovered cfDNA consisted of a broad distribution of fragments ranging from approximately 200 to 6000 base pairs (Figure 2D)."• Some comments here:• The cfDNA size profile is interesting. The short fragments demonstrate an apoptotic laddering pattern, but there is also a population of long cfDNA fragments, which may indicate an origin from apoptosis. Could the authors elaborate on this?

The reviewer raises a valuable point. Accordingly, we have expanded our discussion to include a statement of possible origins of cfDNA fragment lengths and how these might be produced through different modes of cellular death.

“The cfDNA we recovered from cardiac organoids consisted of a heterogeneous distribution of fragment lengths that ranged from approximately 200 to 6000 base pairs. The shorter fragments are potentially indicative of DNA that has been cleaved during apoptosis, while the longer lengths could result from necrotic processes [19]. However, cfDNA can also be expelled actively from healthy cells in the form of extracellular vesicles or nucleoprotein complexes. Further, different DNases may contribute uniquely to patterns of fragmentation [20]. Tracing individual fragments back to the cellular event triggering their release would be a valuable area of future research.”

• A recent publication on the characterization of cfDNA in 2D cell cultures suggested that shorter fragments and longer fragments may originate from the same process and that longer fragments are degraded into shorter fragments over time.

Although we attempted to minimize degradation by storing DNA at -80^o^C, limiting freeze/thaw cycles, and keeping DNA/reagents at 4^o^C during protocols, it is indeed possible that longer fragments could have been degraded into shorter fragments post-collection. To highlight this potential source of error, we have included the following statements in the limitations section of our discussion.

“… although we attempted to minimize DNA degradation post-collection by limiting freeze-thaw cycles and using good sample storage practices, we cannot rule out that longer fragments of DNA could have broken down into shorter ones over time.”

o This paper also shows that the apparently high concentration of longer cfDNA fragments as seen in electropherograms may be an artefact of the method. Could the authors comment on this?

We are unfortunately not completely clear which paper/method the reviewer is referring to. We located this study (doi: 10.2144/btn-2022-0040) which shows that standard culture methods may introduce contaminating cfDNA fragments, resulting from FBS. However, our culture system uses serum-free media so we do not believe this is of concern. In addition, we wish to emphasize that our intention in showing the electropherograms was only to provide a qualitative assessment of fragment lengths, and we agree that definitive conclusions or quantifications cannot be drawn from these. Accordingly, we have included the following statements in our discussion.

“Further, the electropherograms which showed cfDNA size distribution are qualitative, and cannot be taken as a definitive statement of fragment profiles, especially since events such as post-collection degradation could be missed through this analysis.”

• If it is true that longer fragments also contain important biological information, this information could be missed in some cases, for example:o The use of extraction kits that are biased toward the capture of short fragmentso DNA sequencing techniques that are limited to short readso cfDNA sizing using PCR techniques

Although we used standard extraction protocols and made our best effort to control for variability, it is of course possible that bias towards specific fragments could have occurred during the extractions. DNA sequencing or sizing of cfDNA using PCR were not employed in this study, but indeed future studies using such techniques should be aware that longer fragments could potentially be excluded. Accordingly, we have included the following statement in our discussion.

“… although the Maxwell ccfDNA automated extraction kit is one of the better tools currently available for cfDNA extraction [24], we cannot rule out potential bias towards short fragments.”

• The method used for cfDNA sizing is not discussed in the methods and materials section.

We thank the reviewer for pointing this out. We have added the methods for cfDNA size profiling shown below. Also, for clarity we created an additional methods subtitle “Quantification of cfDNA concentration, fragment analysis, and gene expression”.

“Fragment analysis of cfDNA size was performed using the 5200 Fragment Analyzer System, Agilent Inc using the HS Large Fragment 50kb Kit (Agilent, DNF-464-0500).”

• It would have been interesting to not only see the cfDNA size profile on one day but how it changes over the course of incubation. Perhaps the authors can speculate on what insights this may reveal.

We had difficulty obtaining an accurate fragment analysis profile at timepoints prior to day 6. This is likely because total cfDNA concentration was lower at earlier timepoints (Figure 2B). However, with more sensitive assays (ddPCR, QuantiFluor), DNA could be detected and analyzed. Still, we wished to include the results of the cfDNA fragment analysis time course and have added this as Figure 2 —figure supplement 1. We have updated the text in this section of our results to the following.

“The recovered cfDNA from mature cardiac organoids on day 9 consisted of a broad distribution of fragments ranging from approximately 200 to 6000 base pairs (Figure 2D). Fragment distributions were more difficult to resolve at timepoints prior to day 6, likely due to low DNA concentrations and sensitivity limitations (Figure 2 —figure supplement 1).”

5. Another factor that may complicate the characterization of organoid-derived cfDNA:• Literature suggests that natural levels of apoptosis /necrosis increase over the course of organoid growth and development. How does one differentiate between cfDNA fragments originating from "natural" vs "induced" cell death? Is this a confounding factor for analysis?

We felt our best effort to account for this possibility was to compare cfDNA from treated (DOX) condition to cfDNA from the control condition, which would reflect cell death occurring as a “natural” part of organoid generation. DOX treatment could indeed impact many tissue-level processes including pathways of cell death and differentiation. However, we wished to focus this study on the ability of cardiac organoids to release cfDNA that could potentially be predictive of normal vs abnormal (DOX-treated) outcomes. Future studies of released cfDNA in response to specific modes of cell death would be valuable.

Reviewer #2:Silver et al. demonstrated that cell-free DNA can be detected from cultures of cardiac organoids. This subject has not been explored and presents a real interest towards the possible use of organoids to detect markers according to the physiological or pathological evolution of the original tissues. So they tried to assess the possibility of cardiotoxicity of a drug from organoid cultures. The writing of the manuscript is of good quality, concise, and the presentation of the results is clear. However, the manuscript has many flaws including many inaccuracies; and authors should address the following major concerns:1. The analysis of free mitochondrial DNA raises many questions: (i), there is no technical explanation for standardization, calibration, and direct comparison of the quantification of the number of copies of different regions of the mitochondrial genome. The quantification of mitochondrial DNA by the quantitative PCR technique must be determined with caution;

We chose to assess the presence of multiple sequences around the mitochondrial genome (Figure 3A) to determine whether cell-free mitochondrial DNA was present in our samples. Normalization to a housekeeping gene such as β-actin, TBP, or GAPDH is difficult in cfDNA. Unlike tissue-level analyses which examine expression of RNA transcripts converted to complementary DNA (cDNA), cell-free DNA (cfDNA) consists of expelled DNA fragments that are present in the extracellular media. It is currently unclear whether certain fragment sequences might be present more consistently than others. For these reasons, we chose droplet-digital PCR, which delivers more accurate counts and does not necessarily require the rigorous normalization of RT-qPCR. However, we thank the reviewer for raising the point that this could be a possible source of error and have included the following statements in our discussion.

“Unlike tissue-level analyses which examine expression of RNA transcripts converted to complementary DNA (cDNA), cell-free DNA (cfDNA) consists of expelled DNA fragments that are present in the extracellular media. It is currently unclear whether certain fragment sequences might be present more consistently than others, so it is therefore difficult to normalize ddPCR analyses of cfDNA copy number to a housekeeping gene. This is a possible source of error in determining the abundance of specific cfDNA sequences. Future studies of normalization and copy number analyses in free nucleic acids would be valuable towards precise quantitative assessments of cfDNA sequences.”

(ii) the readers may remain confused regarding the method used to determine DNA concentration;

As stated in our Materials and methods section, we determined DNA concentration using the Promega QuantiFluor dsDNA System, which employs a fluorescent double-stranded DNA-binding dye (504nmEx/531nmEm) to enable sensitive quantitation of DNA concentration.

(iii) the reviewers is confused throughout the manuscript in respect to characterize gene expression (RT-PCR) and cfDNA amount (qPCR);

We wish to clarify that we used droplet-digital PCR (ddPCR) for all gene characterizations in this study. To quantify gene expression at the tissue-level, we extracted RNA from the organoids using the Maxwell automated nucleic acid extraction system, then converted the RNA to complementary DNA (cDNA) using the iScript reverse transcription kit, then quantified cDNA using ddPCR with probes specific to genes of choice. In the case of cell-free DNA (cfDNA), we extracted free DNA from conditioned media and used ddPCR to determine copy number.

(iv), the authors state in the abstract that there is a decrease in the fraction of free mitochondrial DNA during development organoids which is not proven by the results;

We thank the reviewer for their careful assessment of our study. However, we respectfully disagree and feel the data indeed show a trending decrease of cell-free mitochondrial DNA as the organoids develop. But to avoid overstating the implications of these data, we have revised our wording in the abstract to the following.

“Intriguingly, we observed that the fraction of cell-free mitochondrial DNA appeared to decrease as the organoids developed, suggesting a possible signature of cardiac organoid maturation, or other cardiac growth-related tissue-level events.”

(v), on what basis can the authors say that this identifies a unique signature of cardiac differentiation if they did not make a comparison with other cell types;

We agree with the reviewer that it is an overstatement to say that these data definitively identify a unique signature of cardiac differentiation. Our intention was to point out the possibility that the observed pattern of decreasing cell-free mitochondrial DNA with growth day could possibly be a signature of cardiac tissue formation. However, many tissue-level events in addition to differentiation (cell death, proliferation) may contribute to cfDNA in our model. Accordingly, we have revised our wording to the following.

“Intriguingly, we observed that the fraction of cell-free mitochondrial DNA appeared to decrease as the organoids developed, suggesting a possible signature of cardiac organoid maturation, or other cardiac growth-related tissue-level events.”

(Figure 3) “Cardiac organoids exhibit a time-dependent decrease in cell-free mitochondrial DNA abundance during growth.”

(vi), the authors do not comment on why the expression of the four mitochondrial regions is equivalent, which is counterintuitive;

We wish to clarify that in our analyses of cell-free DNA (cfDNA) we are not measuring expression (RNA cDNA), rather we used ddPCR to assess the abundance of sequences detectable using probes targeting certain gene regions. As shown in Figure 3A, we are determining the presence of several mitochondrial gene sequences in cfDNA, not transcript expression of mitochondrial genes. We believe that the presence of multiple gene regions within the mitochondrial genome suggests that most of the mitochondrial genome is present in cfDNA, which is interesting as it could indicate preferential retention of cells with more mature mitochondria or increased cfDNA release from stressed mitochondria.

(vii), Figure 3C shows standard variations when it looks impossible when showing ratios; and

In this figure we wished to compare the amounts of the four mitochondrial markers we examined in cell-free DNA, to confirm whether or not they were indeed approximately equally represented in the cell-free DNA. To do this, we took the ratios of each of three markers (ND1, ND4, and ND6) to MT-COX2. If the amounts of ND1 and MT-COX2 were equal for example, the ratio would be 1. To check whether or not this was the case, we performed a one-sample t-test for each marker at each timepoint, comparing to a hypothetical value of 1. We observed that the markers were represented approximately equally in abundance at all timepoints, with respect to MT-COX2. Since there were three biological replicates performed in this experiment, standard deviations would definitely be expected. Although there is variability in these ratios, they are very similar and in most cases do not deviate significantly from a value of 1.

(viii), there is no discussion when comparing amount of cfDNA from both origins while they showed an apparent similar level, especially in regards to the literature comparing for instance cfDNA from mitochondria or nucleus.

In this study we did not make any direct comparisons between total concentrations of nuclear and mitochondrial sources of cfDNA. Care must be taken when evaluating these concentrations as the concentration (copies/µL) of one marker (i.e. NKX2.5, p53), may not necessarily be representative of the total concentration of cfDNA from genomic origin, which can be composed of many different fragments. Our goal was to determine whether cfDNA of both mitochondrial and genomic origin can be detected in cfDNA derived from cardiac organoids, which we feel we were able to successfully demonstrate. We are unfortunately unclear what literature example the author is referring to and hope that we have answered the question appropriately.

2. How is it possible that wild-type DNA, in this case P53, can show a difference in quantity by quantitative PCR analysis during differentiation?

This is what the data have shown us. We wish to clarify that in this experiment, we are not looking at gDNA, nor RNA expression but cell-free DNA (cfDNA), which is very different. CfDNA is released largely from dying cells or enclosed within extracellular vesicles and does not necessarily follow the same dynamics as DNA inside living cells. Epigenetic changes such as histone wrapping or protein-DNA interactions may protect certain regions of DNA from destruction by DNases or degradation. Therefore, changes in the abundance of recovered cfDNA fragment sequences may reflect changing events occurring in the cells from which they originated.

3. No or very few indications specified the number of passages or the number of organoids within the same culture. This is crucial and must be filled in.

This information is provided in our Materials and methods section, as shown below. If the reviewer is referring to the passages of organoids, we wish to clarify that the cardiac organoids grow continuously over the time course and are not individually passaged. In addition, we provide a statement of the number of cardiac organoids cultured per replicate (batch), shown below. To further clarify, we collected conditioned media from appx. 10 organoids per technical replicate, 2 technical replicates per batch, 3 batches total.

“Cells were passaged between 60 and 80% confluence using 0.5 mM EDTA (Gibco, 15575) onto plates pre-coated with growth factor-reduced Matrigel (Corning, 354230) for 30 min at 37oC in DMEM/F:12 (Gibco, 11330 032) at a concentration of 1.2 µL/mLcm2. Cultures were passaged a maximum of 10 times provided no morphological changes indicative of differentiation were present.”

“CfDNA from cardiac organoids was collected from media conditioned by 10 tissues (2-3 technical replicates per batch, 3 independent batches).”

Reviewer #3:The article analyses for the first time the kinetics of different cfDNA species during the development of cardiac organoids generated from H9 human embryogenic stem cells. The stimulation of the cells with Doxorubicin or CPI, as a model of toxic exposure, is a relevant approach to identifying the validity of cfDNA as biomarkers of toxic events. The authors show an increase in cfDNA and a decrease in mtDNA during cardiac organoid differentiation. However, the research relies on a single cell line, and the number of biological replicates with n = 3 is very small. The inclusion of additional two or three replicates is required to validate the finding. Moreover, the inclusion of another selected cell line would strengthen the finding.

We thank the reviewer for raising this important discussion. Many cell culture studies conventionally use three biological replicates per experiment. However, we agree with the reviewer that many studies could benefit from more replicates, as n=3 is the minimum number needed to evaluate statistical significance, and may not resolve smaller changes between conditions. Still, there is debate in this area as some researchers feel that increasing replicate number can lead to “chasing significance” and bias analysis. Due to the time involved in organoid generation (appx. two weeks per batch) and cost, we decided beforehand to perform three separate batches (biological replicates) and evaluate the data afterwards, to limit our bias/expectations of how the cfDNA data should turn out. We wish to further clarify that at least two technical replicates were collected from each batch, using conditioned media collected from a total of appx. 10 organoids. Each cfDNA sample is then representative of the average released cfDNA from all of these organoids, not just one tissue per batch. Still, we wholeheartedly agree that these data should not be taken as a definitive statement of cardiac cfDNA markers in the body in response to DOX or other conditions. Rather, we feel all our experiments combined show the utility of cardiac organoid systems for revealing candidate cfDNA biomarkers that may be reflective of normal growth vs malformation. We feel that exploring additional cell lines and organoid systems would indeed be a valuable next step, but beyond the scope of the current study. Accordingly, we have added the following discussion.

“… our study only assessed organoids derived from one cell type (H9 ESCs) in a limited number of batches (n=3). However, we feel this work has demonstrated that cfDNA can be successfully recovered from cardiac organoids in quantities sufficient for characterization and quantification. This lays the groundwork for further studies of cfDNA from additional organoid systems in response to toxicant treatments.”

The article is well-written and the authors used various methodological approaches to prove their findings. However, with respect to the main message, namely, the increase of specific cfDNA targets in response to DOX treatment, the reviewer has some concerns. Those are related to statistical aspects as well as methodological as described below.– In figures 1-4 the authors compare paired samples over time. However, in their figure legends, the authors indicate that an unpaired test was used. Does this have any rationale like missing samples? The single point of the samples should be included in the figure.

Although within the same batch, the samples compared on different growth days are not necessarily from exactly the same tissues on the plate. For example in Figure 1, tissue-level protein expression was examined by collecting tissues on each growth day. For consistency and unbiased sample analysis, we chose to perform an unpaired t-test with Welch’s correction. The graphs show the average of three biological replicates + SD. We have revised the figure legends to state the method of statistical analysis directly after the graph description, for clarity.

– In all figure legends it is noted that t-tests with Welch´s correction were used unless otherwise indicated. The authors could indicate if a t-test (homogenous variances between samples) or a welch´s t-test (not normal variances) was used.

We wish to clarify that all t-tests performed were unpaired t-tests with Welch’s correction. The exception was in two graphs (3B and 3C), where we wished to compare samples normalized to growth day 1 (3B) or ratios of markers (3C). Since we were comparing the significance of the samples to a hypothetical value of one, we could not use the unpaired t-test with Welch’s correction. Rather, we used a one-sample t-test comparing to a hypothetical value of 1.0.

– The authors should additionally highlight which of the results are related to cfDNA or cDNA analysis. For example, Figures 4 B and C relate to cfDNA (labelled with cfDNA). Figure 5 J relates to cDNA (not labelled). Figure 6 (not labelled with cfDNA).

We thank the reviewer for pointing out that this could be unclear. In addition to the label in the figure legend we have added an additional label “tissue-level expression” above graph 5J.

– Figure 3 B, please clarify if the relative abundance of mtDNA relates to total cfDNA or one of the reference genes.

The relative abundance of mtDNA would definitely contribute to total cfDNA concentration in addition to cfDNA from genomic origin. The same amount of cfDNA (0.5 or 1 ng) was added to each ddPCR reaction.

– Figure 4 should be restructured. Figure 4B, C, F, and G belong together, and D and E belong together. As long as F and G display cfDNA analysis (should be highlighted). Aftewards the legend text could be reduced relevantly.

We thank the reviewer for this suggestion. For clarity, we moved Figure 4 panels F and G to their own supplementary figure (Figure 4 —figure supplement 1), labeled “cfDNA”.

– Figure 6: The copy number of the different cfDNA targets relies relatively on the input material (0.5 or 1 ng were included in the ddPCR assay). Especially for Figure 6 and the discussion that ND1 and NKx2.5 increase in response to DOX treatment, but not the total amount of cfDNA and not p53. Can this result be strengthened by the analysis of GAPDH or TBP in the same sample?

We wish to clarify that the same amount of cfDNA was used in each assay (due to limited sample volume, we decreased the amount to 0.5 ng for some experiments, but ensured the same amount was used for each PCR.) Normalization of cfDNA to a housekeeping gene such as GAPDH or TBP is difficult, as released fragmented DNA sequences may not necessarily be consistently present as with studies of gene expression inside the cell. For this reason, we opted to use ddPCR which provides a more accurate count of copy number using a droplet-based approach and does not necessarily require a housekeeping gene like RT-qPCR. To address this point we have added the following discussion.

“Unlike tissue-level analyses which examine expression of RNA transcripts converted to complementary DNA (cDNA), cell-free DNA (cfDNA) consists of expelled DNA fragments that are present in the extracellular media. It is currently unclear whether certain fragment sequences might be present more consistently than others, so it is therefore difficult to normalize ddPCR analyses of cfDNA copy number to a housekeeping gene. This is a possible source of error in determining the abundance of specific cfDNA sequences. Future studies of normalization and copy number analyses in free nucleic acids would be valuable towards precise quantitative assessments of cfDNA sequences.”

– The reviewer feels that 3 more replicates are needed to validate the finding of different cfDNA release in response to DOX. Moreover, a comparison to another suitable and well-selected cell line would be welcome.

Addressed in the first response above.